# Pain and sociodemographic factors as moderating roles in the relationship between postoperative nausea–vomiting and recovery quality: A process macro modeling study from a nursing perspective on laparoscopic cholecystectomy patients

İlknur Tascı[1¤a], Gizem Kubat Bakır[2¤b]*

**1** Department of Nursing, Fatih Sultan Mehmet Training and Research Hospital, İçerenköy - Ataşehir, Istanbul, **2** Department of Nursing, Maltepe University, Maltepe, Istanbul

¤a Fatih Sultan Mehmet Training and Research Hospital, University of Health Sciences, Icerenköy, D100 Overpass, No: 1, Ataşehir, Istanbul, Türkiye,
¤b Maltepe University, School of Nursing, Buyukbakkalkoy, Maltepe Unv. Marmara Egitim Koyu, Maltepe Istanbul/Turkey
* gzmkbt@gmail.com

## Abstract

Postoperative nausea, vomiting, and pain are common complications following laparoscopic cholecystectomy, and their impact on recovery quality remains a critical concern in surgical nursing practice. The aim of this study was to examine the relationship between postoperative nausea and vomiting and recovery quality in patients undergoing laparoscopic cholecystectomy and to evaluate the moderating roles of sociodemographic factors and pain in this relationship. A cross-sectional, multi-center, correlational design was employed using validated scales and PROCESS Macro (Model 1) moderating analysis to examine factors influencing postoperative recovery among patients undergoing laparoscopic cholecystectomy. This cross-sectional, multi-center, correlational study was conducted between December 2024 and April 2025 in the surgical units of a public training and research hospital and a foundation university hospital in Istanbul, Türkiye, with a sample of 200 patients who underwent laparoscopic cholecystectomy and met the inclusion criteria. Data were collected using the "Descriptive Information Form," the "Quality of Recovery-40 Scale," the "Rhodes Index of Nausea, Vomiting, and Retching," and the "Visual Analog Scale." Descriptive statistics, reliability analyses, confirmatory factor analysis, and moderating analyses were used in data analysis. The mean age of the patients was $52.26 \pm 12.48$ years, and most patients were aged 40 years or older, female, married, non-smokers, and non-alcohol users. The mean BMI was $28.89 \pm 3.21$. More than half of the patients reported no chronic disease, and most

**Data availability statement:** All relevant data are within the paper and its Supporting Information files.

**Funding:** The author(s) received no specific funding for this work.

**Competing interests:** The authors have declared that no competing interests exist.

had a history of previous surgery. The mean VAS pain score was 6.49 ± 0.58, the mean Rhodes Index score was 15.13 ± 0.84, and the mean Quality of Recovery-40 score was 143.18. Among Quality of Recovery-40 subdimensions, the highest mean score was observed in the physical comfort domain (49.70 ± 0.53). Recovery quality was found to be negatively and strongly correlated with nausea and vomiting. The results showed that certain sociodemographic variables (gender, BMI, income level, smoking status, presence of chronic illness) and pain had significant moderating g effects in the relationship between nausea and vomiting and recovery quality, with pain having the strongest effect. Age, marital status, education level, alcohol use, and type of surgery did not show significant interaction effects. This study demonstrated that postoperative symptoms such as nausea, vomiting, and pain significantly affect recovery quality in patients undergoing laparoscopic cholecystectomy, and that this relationship is mediated by specific sociodemographic factors. These findings highlight the need for enhanced symptom management and individualized care approaches in nursing practice to improve postoperative recovery outcomes.

## 1. Introduction

The postoperative period is a critical phase in the continuum of surgical care, where nurses play a pivotal role in promoting recovery, managing symptoms, and improving patient outcomes. With the global transition toward minimally invasive techniques, laparoscopic cholecystectomy has become the gold standard for the treatment of gallbladder diseases. While laparoscopic cholecystectomy offers numerous clinical benefits such as reduced postoperative pain, shorter hospital stay, and faster return to daily activities patients often continue to experience distressing symptoms such as pain, nausea, and vomiting. These symptoms, if poorly managed, can significantly impair recovery quality and patient satisfaction, both of which are key indicators of high-quality nursing care [1–6]

Nurses are uniquely positioned to assess and address postoperative discomfort through timely interventions, patient education, and individualized symptom management. However, there remains a critical gap in understanding how postoperative nausea and vomiting and pain interact with patient-specific characteristics to influence recovery outcomes. Moreover, the concept of recovery quality is increasingly viewed not only through the lens of physical healing but also through emotional well-being, functional independence, and perceived support dimensions that are highly responsive to nursing interventions [6,9,11,12].

This study was designed to evaluate the moderating effects of postoperative pain and sociodemographic variables in the relationship between postoperative nausea and vomiting and perceived recovery quality among patients undergoing laparoscopic cholecystectomy. Findings from this research are expected to inform evidence-based nursing strategies aimed at symptom control, early recovery optimization, and patient-centered perioperative care.

## 1.1. Background

Gallbladder diseases ranging from cholelithiasis and cholecystitis to neoplasms are among the most common gastrointestinal disorders globally, often requiring surgical intervention [1–3]. Laparoscopic cholecystectomy has largely replaced open surgery due to its minimally invasive nature and superior recovery profile3. While laparoscopic cholecystectomy offers clear advantages, it is not without complications. Despite reduced incision size and faster mobilization, patients frequently report symptoms such as postoperative nausea and vomiting and postoperative pain, both of which may compromise the recovery experience [4,5]

Nausea and vomiting following laparoscopic cholecystectomy are multifactorial in origin. Pneumoperitoneum with $CO_2$, patient positioningreverse Trendelenburg, and delayed gastric emptying may all contribute to increased postoperative nausea and vomiting risk. Hypercapnia-induced stimulation of the medullary vomiting center and vestibular system further exacerbates the problem [6,7]. From a nursing standpoint, the effective prevention and control of postoperative nausea and vomiting is essential to enhancing postoperative comfort and recovery.

Postoperative pain is another significant challenge, even in laparoscopic procedures. Pain arises from peritoneal stretching, diaphragmatic irritation, and tissue trauma. Patients often report somatic pain at trocar sites and referred shoulder pain due to $CO_2$ retention and phrenic nerve stimulation [8,9]. Severe pain not only prolongs recovery but also undermines patient trust in care and increases reliance on pharmacological interventions. Timely, evidence-based nursing assessment and intervention are critical in mitigating these adverse outcomes [5].

Recovery quality is a multidimensional outcome encompassing physical comfort, emotional well-being, functional mobility, and patient-perceived support. It is now recognized as a key quality indicator in nursing care delivery [10]. The Quality of Recovery-40 (QoR-40) tool, among others, allows for structured evaluation of patient-reported recovery outcomes12. Research indicates that postoperative nausea and vomiting, severe pain (VAS ≥ 7), longer surgical duration, and existing comorbidities significantly impair recovery quality [11]. Furthermore, individuals experiencing high postoperative symptom burden are at a disproportionately higher risk of poor outcomes.

The early identification of patients at risk for poor recovery and the implementation of individualized nursing care plans are essential components of high-quality perioperative nursing. Nurses are not only care providers but also coordinators of multidisciplinary interventions, educators, and advocates for patient well-being. Understanding how postoperative nausea and vomiting and pain intersect with sociodemographic variables to influence recovery is vital for tailoring interventions and advancing the science of recovery-oriented nursing practice [13].

This study contributes to the literature by providing a clearer understanding of how postoperative nausea and vomiting are associated with recovery quality in patients undergoing laparoscopic cholecystectomy within a moderating framework. By examining the role of pain and selected sociodemographic characteristics in shaping this relationship, the study adds nuance to existing evidence that has largely focused on direct associations. The findings offer practical implications for postoperative nursing care by supporting more individualized symptom management approaches and by informing patient-centered planning of postoperative care.

The research questions of this study were formulated to address clinically relevant gaps in understanding postoperative recovery after laparoscopic cholecystectomy. Specifically, focusing on the relationship between postoperative nausea–vomiting and recovery quality, and examining the moderating role of pain and selected sociodemographic characteristics, allows for a more comprehensive understanding of recovery processes. This approach enhances the relevance of the study by linking empirically testable questions to clinically meaningful outcomes in postoperative nursing care.

## 2. Materials and methods

### 2.1. Research model and aim

In this study, a quantitative and correlational research design was employed to examine the direct and indirect relationships among variables. Confirmatory factor analyses were conducted using LISREL. Hypotheses were tested using PROCESS

Macro (Model 1), a regression-based moderation approach. Given the cross-sectional design, causal inferences cannot be made from the observed associations. The PROCESS Macro was used because it provides a robust and widely accepted framework for testing moderating effects in observational data through regression-based interaction models.

The aim of this study was to investigate the relationship between postoperative nausea and vomiting and recovery quality in patients undergoing laparoscopic cholecystectomy, and to evaluate the moderating role of sociodemographic characteristics and postoperative pain in this relationship.

### 2.2. Research questions

The research questions guiding this study are as follows:

- What are the sociodemographic and health history characteristics of patients who have undergone laparoscopic cholecystectomy?

- What is the level of recovery quality among patients who have undergone laparoscopic cholecystectomy?

- What are the levels of nausea, vomiting, and retching as measured by the Rhodes Index among patients who have undergone laparoscopic cholecystectomy?

- What are the pain levels of patients who have undergone laparoscopic cholecystectomy as measured by the Visual Analog Scale?

- Is there a relationship between recovery quality and the levels of nausea, vomiting, and pain in patients who have undergone laparoscopic cholecystectomy?

- Do sociodemographic characteristics and pain levels have a moderating role in the relationship between nausea, vomiting, and retching and recovery quality in patients who have undergone laparoscopic cholecystectomy?

### 2.3. Study Setting and Duration

This study was conducted between December 2024 and April 2025 in the surgical units of a public training and research hospital and a university-affiliated medical faculty hospital located in Istanbul, Türkiye. The patient population consisted of individuals who underwent laparoscopic cholecystectomy during the specified time period.

### 2.4. Population and Sample

The study population comprised patients who underwent laparoscopic cholecystectomy between December 2024 and April 2025 in the surgical units of a public training and research hospital and a university-affiliated medical faculty hospital in Istanbul, Türkiye. The study sample consisted of patients who met the inclusion criteria and underwent laparoscopic cholecystectomy during this period.

The required sample size was calculated using G*Power software. To determine the input parameters for the power analysis, relevant literature was reviewed. An a priori power analysis was conducted based on a two-tailed test, a medium effect size (d = 0.5), a 5% alpha error probability, and a statistical power of 80% (1-β). Based on this analysis, a minimum of 128 patients was determined to be sufficient for the study.

### 2.5. Inclusion and exclusion criteria

Inclusion criteria:

- Patients were eligible to participate in the study if they:

- Were over the age of 18,

- Were oriented to time and place,

- Had no hearing or visual impairments,

- Were literate,

- Were able to speak Turkish,

- Underwent laparoscopic cholecystectomy under general anesthesia,

- Voluntarily agreed to participate in the study and provided written informed consent,

- Were not admitted to the intensive care unit postoperatively, and

- Had no diagnosed psychiatric illness.

**Exclusion criteria:**

- Patients were excluded from the study if they:

- Were initially scheduled for laparoscopic cholecystectomy but converted to open cholecystectomy intraoperatively,

- Underwent laparoscopic cholecystectomy with spinal anesthesia,

- Declined to participate in the study,

- Were unable to communicate verbally,

- Had hearing loss or any condition preventing data collection, or

- Withdrew from the study voluntarily.

### 2.6. Data collection and instruments

Four data collection tools were used in this study: the "Descriptive Information Form," the "Visual Analog Scale (VAS)," the "Quality of Recovery-40 (QoR-40) Questionnaire," and the "Rhodes Index of Nausea, Vomiting, and Retching." All data were collected through face-to-face interviews within the first 24 hours following surgery.

**2.6.1. Descriptive Information Form.** This form, developed based on a review of the relevant literature [14,15], was used to collect sociodemographic and clinical characteristics of the patients who voluntarily participated in the study. The form includes a total of 11 items: 9 questions assessing age, gender, height, weight, marital status, educational level, income status, smoking and alcohol use, and 2 questions evaluating chronic disease status and previous surgical history.

**2.6.2. Visual Analog Scale (VAS).** The Visual Analog Scale, developed by Price and colleagues [16], is a widely used tool for evaluating pain intensity. The scale consists of a 10 cm horizontal line marked from 0 to 10, with each point representing a specific pain level: 0 = no pain, 1–4 = mild pain, 5–6 = moderate pain, 7–8 = severe pain, and 10 = unbearable pain. The score selected by the patient reflects the severity of pain experienced. The VAS has demonstrated high validity and reliability in clinical practice.

**2.6.3. Quality of Recovery-40 (QoR-40) Scale.** The QoR-40 was developed by Myles et al. (2000) to assess patients' recovery following surgery and anesthesia [17]. The Turkish adaptation and psychometric validation of the instrument were conducted by Karaman et al. (2014) [18]. The scale includes 40 items grouped into five subdimensions: physical comfort, emotional state, physical independence, patient support, and pain. Items are rated on a 5-point Likert scale. Section A includes positively worded items scored from 1 ("none of the time") to 5 ("all of the time"), whereas section B includes negatively worded items scored in reverse. The total score ranges from 40 to

200, with higher scores indicating better recovery. In the current study, the Cronbach's alpha coefficient was found to be 0.90.

**2.6.4. Rhodes Index of Nausea, Vomiting, and Retching (INVR).** Originally developed by Rhodes and McDaniel (1999), the Rhodes Index is designed to assess the frequency and distress associated with postoperative nausea, vomiting, and retching [19]. The Turkish adaptation was validated in 2010 (Genç, 2010) [20]. The scale consists of eight items, with items 1, 3, 6, and 7 reverse-scored. Each item is rated from 0 (none) to 4 (most), and the total score ranges from 0 to 32, with higher scores indicating worse nausea–vomiting experience. In the present study, the Cronbach's alpha coefficient was calculated as 0.94.

## 2.7. Data analysis and interpretation

Descriptive statistics were first conducted. Categorical variables were presented as frequencies and percentages, while continuous variables were expressed as mean ± standard deviation (SD). The construct validity of the scales was tested through Confirmatory Factor Analysis (CFA) using LISREL 8.80. Model fit was evaluated based on chi-square/degree of freedom, RMSEA, SRMR, CFI, and NNFI indices. An RMSEA value below 0.08 indicated an acceptable model fit. Relationships among the variables were examined using Pearson's correlation analysis. According to the $r$ value, correlations below 0.30 were considered weak, between 0.30–0.70 moderate, and above 0.70 strong. Hayes' PROCESS Macro (Model 1) was employed to test the hypotheses [21]. In this model, daily living activities were treated as the independent variable and postoperative recovery as the dependent variable. BMI, education, income, smoking, alcohol consumption, and previous surgery were tested as moderators. The bootstrap technique with 5000 resamples was applied, and 95% confidence intervals were calculated.

Beta coefficients and confidence interval (CI) values were reported. All analyses were performed using IBM SPSS Statistics version 26.0. The significance level was set at $p < 0.05$, and results with $p < 0.01$ were additionally indicated. $p$ values were reported with two decimal places, and three decimals were used when $p < 0.01$.

## 2.8. Ethical considerations

Ethical approval was obtained from the Scientific Research and Publication Ethics Committee of a foundation university (decision dated 17 October 2024; No. 2024/19–03) prior to the commencement of the study. Official permission was granted by the Istanbul Provincial Directorate of Health (dated 13 December 2024; No. 2024/18), and institutional approvals were subsequently obtained from the participating hospitals (No. E-17073117-050.99-260953282 and No. E-32835138-108.01-333454). Written informed consent was obtained from all participants prior to data collection. Permission to use the Quality of Recovery-40 (QoR-40) scale and the Rhodes Index of Nausea, Vomiting, and Retching was obtained from the respective authors. For confidentiality purposes, personal data were anonymized, coded, and stored on encrypted computers accessible only to the research team. The study was conducted in accordance with the Declaration of Helsinki and Good Clinical Practice guidelines.

## 3. Results

Table 1 presents the findings related to the patients' sociodemographic characteristics and health history.

Table 1 provides the descriptive statistical findings regarding the sociodemographic characteristics of the study participants. The majority were aged 40 years and older (60.5%), female (63.5%), and married (88.5%). Most participants were classified as overweight or obese (78%), had an educational attainment of high school or below (75%), and reported an income level equal to their expenditures (52.5%). Additionally, 35% of the participants were smokers, 5.5% reported alcohol consumption, and 46% had at least one chronic disease. A substantial proportion had undergone a previous surgical procedure (75.5%). Notably, a high proportion of participants presented with elevated postoperative pain levels, as indicated by VAS scores of ≥6 (60.5%).

**Table 1. Sociodemographic Characteristics and Health History of the Patients.**

| Age Distribution | *n* | % | *M* | *SD* |
|---|---|---|---|---|
| 23–29 years | 8 | 4.0 | 52.26 | 12.484 |
| 30–39 years | 71 | 35.5 | | |
| 40 years and above | 121 | 60.5 | | |
| **Gender** | | | | |
| Female | 127 | 63.5 | – | 1.362 |
| Male | 73 | 36.5 | | |
| **Height Distribution** | | | | |
| 115–164 cm | 83 | 41.5 | 165.52 | 9.609 |
| 165 cm and above | 117 | 58.5 | | |
| **Weight Distribution** | | | | |
| 53–74 kg | 84 | 42.0 | 78.88 | 14.516 |
| 75–94 kg | 85 | 42.5 | | |
| 95 kg and above | 31 | 15.5 | | |
| **Body Mass Index (BMI)** | | | | |
| Underweight (BMI < 18.5) | 2 | 1.0 | 28.89 | 3.214 |
| Normal weight (18.5 ≤ BMI < 25) | 42 | 21.0 | | |
| Overweight (25 ≤ BMI < 30) | 85 | 42.5 | | |
| Obese (BMI ≥ 30) | 71 | 35.5 | | |
| **Marital Status** | | | | |
| Married | 177 | 88.5 | – | 0.319 |
| Single | 23 | 11.5 | | |
| **Educational Status** | | | | |
| Primary school graduate | 79 | 39.5 | – | 0.772 |
| Secondary school graduate | 22 | 11.0 | | |
| High school graduate | 49 | 24.5 | | |
| University graduate | 41 | 20.5 | | |
| Postgraduate | 9 | 4.5 | | |
| **Income Level** | | | | |
| Income less than expenses | 87 | 43.5 | – | 0.566 |
| Income equal to expenses | 105 | 52.5 | | |
| Income greater than expenses | 8 | 4.0 | | |
| **Smoking Status** | | | | |
| Yes | 70 | 35.0 | | 0.478 |
| No | 130 | 65.0 | | |
| **Daily Cigarette Consumption** | | | | |
| 1–5 cigarettes | 15 | 7.5 | – | 0.835 |
| 6–10 cigarettes | 10 | 5.0 | | |
| 11 or more cigarettes | 42 | 21.0 | | |
| **Alcohol Use** | | | | |
| Yes | 11 | 5.5 | – | 0.228 |
| No | 189 | 94.5 | | |
| **Presence of Chronic Disease** | | | | |
| Yes | 92 | 46.0 | – | 0.431 |
| No | 108 | 54.0 | | |

*(Continued)*

**Table 1.** (Continued)

| Age Distribution | *n* | % | *M* | *SD* |
|---|---|---|---|---|
| **History of Previous Surgery** | | | | |
| Yes | 151 | 75.5 | – | 2.093 |
| No | 49 | 24.5 | | |
| **VAS Score** | | | | |
| 0–2 range | 9 | 4.5 | 6.49 | *0.5831* |
| 3–6 range | 70 | 35.0 | | |
| 6 and above | 121 | 60.5 | | |

Note. n = number of patients; % = percentage; M = mean; SD = standard deviation; BMI = body mass index; VAS = Visual Analog Scale.

### 3.1. Findings of the First-Order Confirmatory Factor Analysis of the Quality of Recovery Scale

The first-order confirmatory factor analysis of the Quality of Recovery Scale was conducted in two stages. In the first stage, particular attention was paid to whether the t-values and standardized factor loadings of the items in the multifactorial model were statistically significant. In line with the literature, t-values above 1.96 are considered significant, while standardized factor loadings are expected to be greater than 0.30. The first-stage confirmatory factor analysis results (t-values) are presented in Fig 1.

To evaluate the construct validity of the Quality of Recovery Scale, a first-order Confirmatory Factor Analysis was conducted. In the initial stage, *t*-values for each item were examined, and six items that did not reach statistical significance ($t < 1.96$) were removed from the model. In the second stage, a final model was established with the remaining items, and standardized factor loadings along with overall model validity were calculated. The Confirmatory Factor Analysis results demonstrated that most items had factor loadings above .40 across the five subdimensions of the scale (Physical Comfort, Emotional State, Independence, Patient Support, and Pain). This finding indicates that the items adequately represented their respective factors. While loadings above .30 are generally considered acceptable, those exceeding .40 are recognized as providing strong structural representation (Fig 2).

Notably, some items within the "Independence" and "Pain" subdimensions displayed relatively low or negative factor loadings. These results were interpreted with careful consideration of the theoretical framework and the overall model fit (Brown, 2015). The overall model fit indices were as follows: $\chi^2/df = 2.100$; RMSEA = 0.079; GFI = 0.90; CFI = 0.91; NFI = 0.92; and SRMR = 0.076. These values fall within the acceptable thresholds reported in the Confirmatory Factor Analysis literature, suggesting that the model demonstrated a good fit with the data (Table 2).

Additionally, descriptive statistics including means, standard deviations, skewness, and kurtosis were examined. Item means ranged from 2.41 to 4.94, and standard deviations were generally between 0.3 and 1.3, indicating meaningful variance among the sample. Skewness and kurtosis values were within the ± 2 range, supporting the assumption of approximate normality and the appropriateness of Confirmatory Factor Analysis.

In conclusion, after the removal of six items, the five-factor structure of the Quality of Recovery Scale was confirmed, demonstrating structural validity for use in this study.

### 3.2. First-Order Confirmatory Factor Analysis Findings of the Rhodes Index of Nausea, Vomiting, and Retching

In this study, a first-order Confirmatory Factor Analysis was conducted to evaluate the construct validity of the Rhodes Index of Nausea, Vomiting, and Retching. Confirmatory Factor Analysis is an advanced analytical technique designed to assess the extent to which a theoretically predetermined measurement structure fits the observed data. The scale's unidimensional structure, consisting of eight items, was examined accordingly. Items 1, 3, 6, and 7 are reverse-coded, and these items were appropriately recoded within the dataset prior to inclusion in the model.

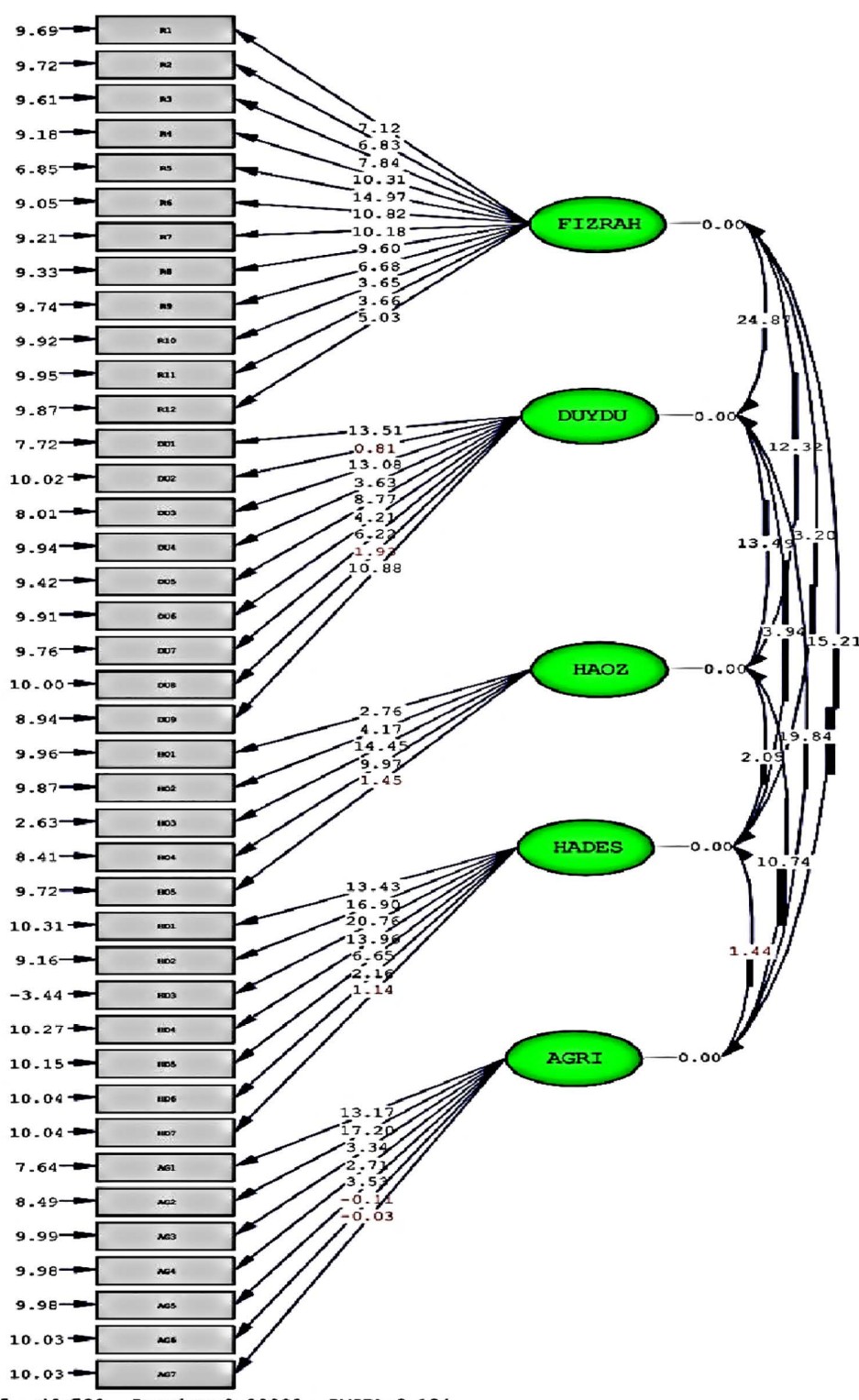

**Fig 1. QoR-40 CFA t-values (Stage 1).** T-values from the first-stage first-order confirmatory factor analysis of the Quality of Recovery-40 (QoR-40) scale. Items with t-values < 1.96 were treated as non-significant and removed in the revised model. Abbreviations: FIZRAH = Physical comfort; DUYDU = Emotional state; HAOZ = Independence in mobility; HADES = Patient support; Pain status, AGRI = PainAs shown in Fig 1, some items

demonstrated t-values below the threshold of 1.96. These items were excluded from the model, and the analysis was repeated. The revised model was evaluated as the second stage of the first-order confirmatory factor analysis. The standardized factor loadings of the items in this revised model are presented in Fig 2. The overall results of the first-order confirmatory factor analysis, including additional fit indices, are summarized in Table 2.

The Confirmatory Factor Analysis was performed using LISREL 8.80 software. Within the analysis, standardized factor loadings, *t*-values, and overall model fit indices ($\chi^2$/df, RMSEA, CFI, GFI, SRMR) were calculated. In addition, descriptive statistics for each item, including mean (M), standard deviation (SD), skewness, and kurtosis, were assessed to examine the distributional properties of the data.

Interpretation of the Confirmatory Factor Analysis results emphasized that factor loadings ≥ .40 indicate strong representation of the underlying construct, while model fit indices meeting the recommended cut-off values provide robust evidence of construct validity.

As presented in Fig 3, the first-order Confirmatory Factor Analysis results of the unidimensional, eight-item Rhodes Index of Nausea, Vomiting, and Retching (*t*-values) are displayed. Since all item *t*-values exceeded the threshold of 1.96, no items were removed from the model; however, three model modifications were applied. In this revised form, the instrument was adapted to the study sample while preserving its original theoretical structure. The detailed findings of the scale and the results of the first-order Confirmatory Factor Analysis are summarized in Table 3.

Table 3 presents the findings of the first-order confirmatory factor analysis conducted to assess the construct validity of the Rhodes Index of Nausea, Vomiting, and Retching. The results demonstrate that the single-factor, eight-item structure is both statistically significant and valid. The analysis was performed using the LISREL 8.80 software, and all item *t*-values exceeded 1.96, indicating significant contributions to the model. The standardized factor loadings of the items ranged from .50 to .97. In the literature, factor loadings above .40 are generally considered sufficient to represent the underlying construct. Accordingly, all items were found to load significantly and strongly onto the factor.

Model fit was evaluated using several fit indices, yielding the following results: $\chi^2$/df = 3.90 ($\chi^2$ = 144.48/ df = 37), RMSEA = 0.064, GFI = 0.90, CFI = 0.93, NFI = 0.91, and SRMR = 0.69. All indices fall within the acceptable thresholds, indicating that the model demonstrated a good fit with the observed data (Byrne, 2016; Schermelleh-Engel et al., 2003). Notably, the RMSEA value below 0.08 and the CFI and GFI values above 0.90 provide strong evidence supporting the validity of the model.

In addition, item-level descriptive statistics, including mean (M), standard deviation (SD), skewness, and kurtosis values, were examined. The item means ranged from 1.40 to 2.16, with SD values between 0.68 and 1.10, indicating adequate variability within the sample. Furthermore, skewness and kurtosis values for all items were within the ± 2 threshold, suggesting an approximately normal distribution and confirming the suitability of the data for confirmatory factor analysis.

### 3.3. Pearson correlation analysis of the scales

In Table 4, the relationships between the Quality of Recovery Scale and its subdimensions and the Rhodes Index of Nausea, Vomiting, and Retching were examined using Pearson correlation analysis; in addition, internal consistency reliability coefficients (Cronbach's alpha) for each subdimension and the overall scales were reported.

According to the correlation analysis, statistically significant relationships were observed between all subdimensions of the Quality of Recovery Scale and the Rhodes Index of Nausea, Vomiting, and Retching (p < .01, p < .05). The mean score of the Quality of Recovery Scale was calculated as 143.18 (SD = 0.47). Within the scale, a minimum score of 40 indicates "very poor" recovery, while the maximum score of 200 reflects "excellent" recovery quality. Accordingly, the total score obtained suggests that patients in the sample generally experienced a moderate to high level of recovery quality. At the subscale level, the highest mean score was observed in *Physical Comfort* (M = 49.70, SD = 0.53), followed by *Patient Support* (M = 28.82, SD = 0.39), *Emotional State* (M = 28.60, SD = 0.59), and *Independence* (M = 16.78, SD = 1.15). The

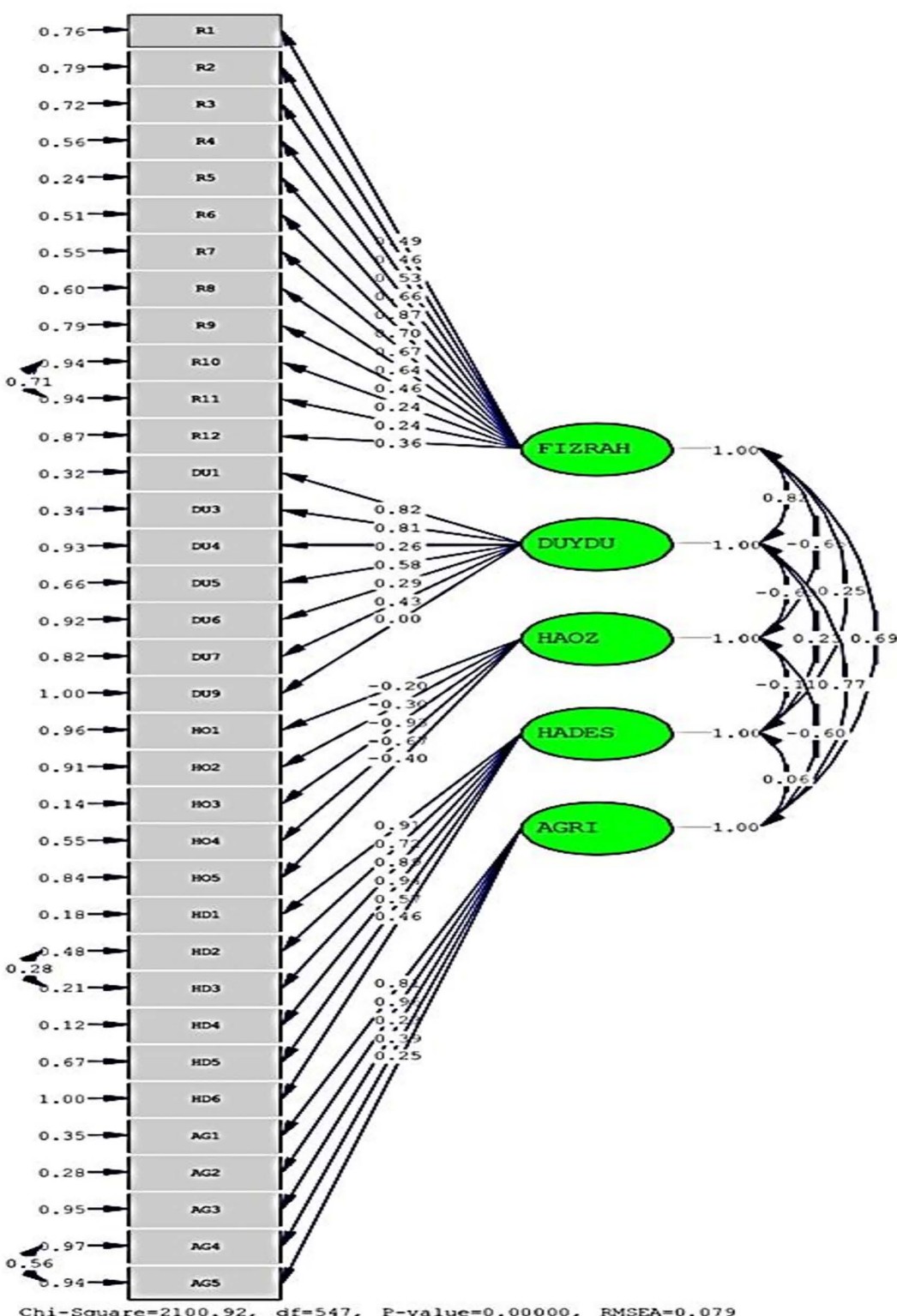

**Fig 2. QoR-40 CFA standardized factor loadings (Stage 2).** Standardized factor loadings from the revised first-order confirmatory factor analysis model of the Quality of Recovery-40 (QoR-40) scale after removal of non-significant items. Abbreviations: FIZRAH = Physical comfort; DUYDU = Emotional state; HAOZ = Independence in mobility; HADES = Patient support; Pain status, AGRI = Pain.

**Table 2. Item-level factor loadings and model fit indices from the first-order confirmatory factor analysis of the Quality of Recovery (QoR-40) scale.**

| Dimensions | Items | Standardized Factor Loadings | t-values | *M (Mean)* | SD (Standard Deviation) | Kurtosis | Skewness |
|---|---|---|---|---|---|---|---|
| **Physical comfort** | R1 | 0.49 | 7.03 | 3.86 | 1.299 | −.903 | −.413 |
| | R2 | 0.46 | 6.54 | 3.03 | 1.166 | .104 | −.997 |
| | R3 | 0.53 | 7.73 | 3.99 | 0.871 | −.866 | .775 |
| | R4 | 0.66 | 10.18 | 2.98 | 1.127 | .008 | −.812 |
| | R5 | 0.87 | 15.05 | 3.88 | 1.139 | −.843 | −.073 |
| | R6 | 0.70 | 10.94 | 4.63 | 0.673 | −1.996 | −0.826 |
| | R7 | 0.67 | 10.33 | 4.58 | 0.703 | −1.660 | 2.106 |
| | R8 | 0.64 | 9.64 | 4.13 | 0.963 | −.911 | −.154 |
| | R9 | 0.46 | 6.59 | 4.83 | 0.591 | −0.922 | −1.018 |
| | R10 | 0.24 | 3.31 | 4.58 | 0.726 | −0.212 | 0.705 |
| | R11 | 0.24 | 3.31 | 4.51 | 0.736 | −0.615 | 0.392 |
| | R12 | 0.36 | 5.01 | 4.66 | 0.653 | −0.033 | 0.864 |
| **Emotional states** | DU1 | 0.82 | 13.50 | 3.05 | 1.087 | 0.420 | −0.934 |
| | *DU2* | *0.08* | *0.81* | *4.64* | *0.687* | *−0.109* | *0.299* |
| | DU3 | 0.81 | 13.28 | 3.17 | 1.144 | 0.139 | −0.098 |
| | DU4 | 0.26 | 3.47 | 4.93 | 0.354 | −0.819 | 0.559 |
| | DU5 | 0.58 | 8.55 | 4.13 | 1.085 | −0.070 | 0.063 |
| | DU6 | 0.29 | 3.96 | 4.94 | 0.377 | −0.502 | 0.510 |
| | DU7 | 0.43 | 5.98 | 4.78 | 0.611 | −0.310 | 0.323 |
| | *DU8* | *0.01* | *1.93* | *4.89* | *0.495* | *−0.493* | *0.492* |
| | DU9 | 0.46 | 10.88 | 3.58 | 1.166 | −0.313 | −1.925 |
| **Independence in mobility** | HO1 | −0.20 | 2.70 | 4.70 | 0.584 | −0.272 | 0.019 |
| | HO2 | −0.30 | 4.12 | 4.20 | 3.701 | 0.108 | 0.767 |
| | HO3 | −0.93 | 14.62 | 3.87 | 0.934 | −0.456 | −0.658 |
| | HO4 | −0.67 | 9.86 | 4.00 | 0.913 | −0.680 | −0.292 |
| | *HO5* | *−0.40* | *1.45* | *2.76* | *0.903* | *0.113* | *0.469* |
| **Patient support** | HD1 | 0.91 | 16.47 | 4.82 | 0.434 | −0.392 | 0.218 |
| | HD2 | 0.72 | 11.44 | 4.75 | 0.553 | −0.549 | 0.031 |
| | HD3 | 0.89 | 15.78 | 4.78 | 0.458 | −0.011 | 0.335 |
| | HD4 | 0.94 | 17.39 | 4.83 | 0.422 | −0.588 | 0.304 |
| | HD5 | 0.57 | 8.62 | 4.90 | 0.317 | −0.154 | 0.701 |
| | HD6 | 0.46 | 3.16 | 4.89 | 0.457 | −0.890 | 0.480 |
| | *HD7* | *0.24* | *1.14* | *4.01* | *0.257* | *−0.914* | *−1.365* |
| **Pain status** | AG1 | 0.81 | 13.16 | 2.41 | 1.094 | 0.579 | −0.364 |
| | AG2 | 0.96 | 16.87 | 3.34 | 1.171 | 0.0314 | −0.026 |
| | AG3 | 0.23 | 3.20 | 4.55 | 0.806 | −0.426 | 0.894 |
| | AG4 | 0.39 | 2.56 | 4.52 | 0.702 | −0.672 | 0.598 |
| | AG5 | 0.25 | 3.42 | 4.33 | 0.947 | −0.648 | 0.615 |
| | *AG6* | *0.02* | *−0.11* | *4.80* | *0.631* | *−0.292* | *0.718* |
| | *AG7* | *0.01* | *−0.03* | *4.90* | *0.536* | *−0.408* | *0.469* |
| | Model Fit Indices | χ²/df = 2.100.92/ 547; RMSEA = 0.079; GFI = 0.90; CFI = 0.91; NFI = 0.92; SRMR = 0.076 | | | | | |

**Note.** M = mean; SD = standard deviation; χ²/df = chi-square/degrees of freedom; RMSEA = root mean square error of approximation; GFI = goodness-of-fit index; CFI = comparative fit index; NFI = normed fit index; SRMR = standardized root mean square residual. All factor loadings and t-values are based on the final CFA model.

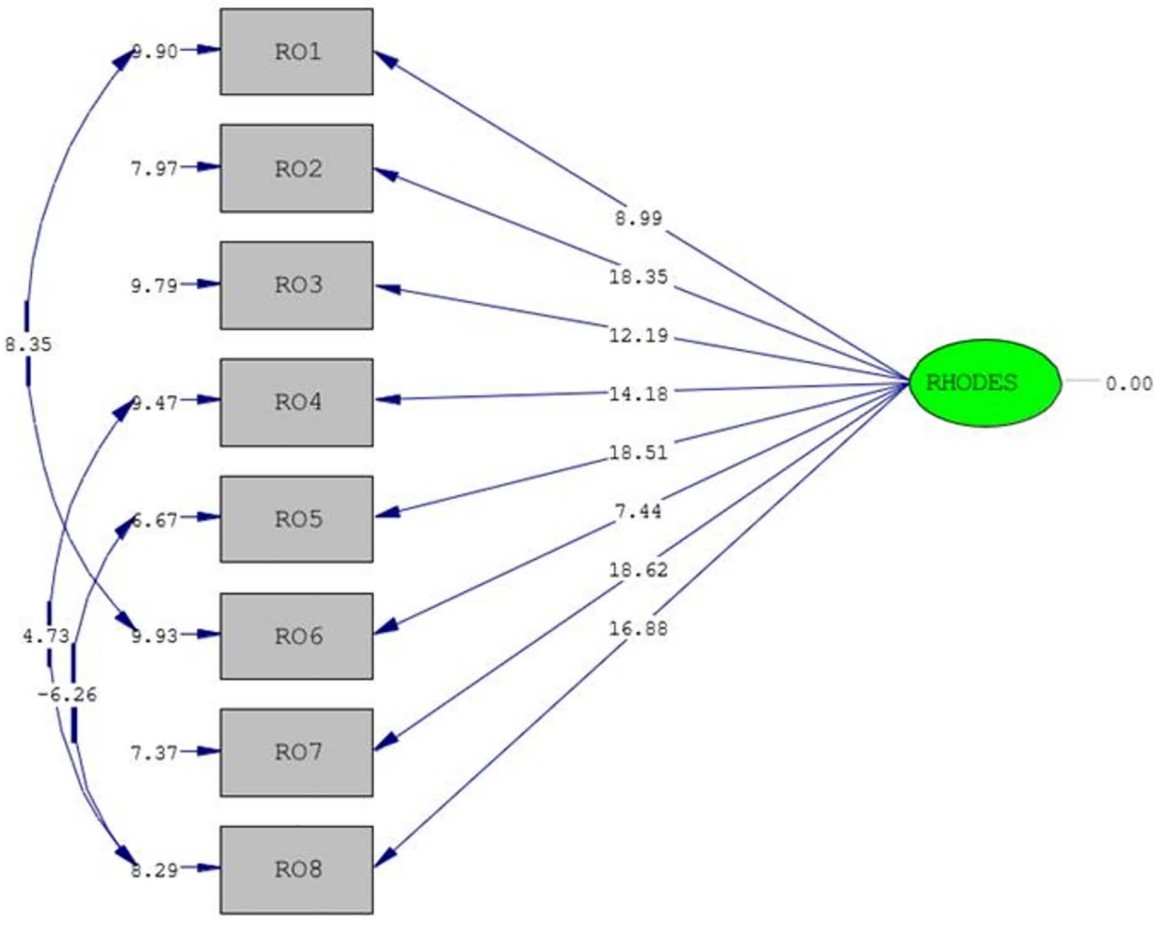

Chi-Square=144.48, df=37, P-value=0.00000, RMSEA=0.064

**Fig 3. Rhodes Index CFA t-values.** T-values from confirmatory factor analysis of the unidimensional eight-item Rhodes Index of Nausea, Vomiting, and Retching. All items showed statistically significant loadings (t > 1.96).

**Table 3. Item-level factor loadings and model fit indices from the first-order confirmatory factor analysis of the Rhodes Index of Nausea, Vomiting, and Retching.**

| Dimensions | Items | Standardized Factor Loadings | t-values | *M (Mean)* | SD (Standard Deviation) | Kurtosis |
|---|---|---|---|---|---|---|
| RO1 | 0.58 | 8.99 | 1.4050 | 0.680 | 1.605 | 1.861 |
| RO2 | 0.96 | 18.35 | 2.1400 | 1.077 | 0.673 | −0.231 |
| RO3 | 0.74 | 12.19 | 1.7000 | 1.041 | 1.355 | 0.901 |
| RO4 | 0.82 | 14.18 | 2.1350 | 1.082 | 0.568 | −0.776 |
| RO5 | 0.96 | 18.51 | 2.1650 | 1.106 | 0.703 | −0.179 |
| RO6 | 0.50 | 7.44 | 1.4000 | 0.722 | 0.967 | 1.568 |
| RO7 | 0.97 | 18.62 | 2.13000 | 1.0385 | 0.607 | −0.345 |
| RO8 | 0.92 | 16.88 | 2.0550 | 1.008 | 0.602 | −0.499 |
| Model Fit Indices | $\chi^2$/df = 144.48/ 37; RMSEA = 0.064; GFI = 0.90; CFI = 0.93; NFI = 0.91; SRMR = 0.69 | | | | | |

M = mean; SD = standard deviation; $\chi^2$/df = chi-square/degrees of freedom; RMSEA = root mean square error of approximation; GFI = goodness-of-fit index; CFI = comparative fit index; NFI = normed fit index; SRMR = standardized root mean square residual.

**Table 4. Pearson correlation coefficients and internal consistency (Cronbach's alpha) of the Quality of Recovery (QoR-40) and Rhodes Index scores.**

| Variables | Total | Mean (X) | *SD* (Standard Deviation) | Quality of Recovery | Physical Comfort | Emotional State | Independence in mobility | Patient Support | Pain Status | RHODES |
|---|---|---|---|---|---|---|---|---|---|---|
| **Quality of Recovery** | 143.18 | 4.21 | .473 | (0.90) | | | | | | |
| **Physical Comfort** | 49.70 | 4.14 | .533 | 0.811** | (0.83) | | | | | |
| **Emotional State** | 28.60 | 4.08 | .587 | 0.811** | 0.756** | (0.78) | | | | |
| **Independence in mobility** | 16.78 | 4.19 | 1.145 | 0.767** | 0.421** | 0.396** | (0.60) | | | |
| **Patient Support** | 28.82 | 4.80 | .391 | 0.358** | 0.237** | 0.233** | 0.103* | (0.87) | | |
| **Pain Status** | 19.17 | 3.83 | .626 | 0.699** | 0.582** | 0.610** | 0.272** | 0.116* | (0.66) | |
| **RHODES** | 15.13 | 1.89 | .837 | −0.635** | −0.777** | −0.619** | −0.303* | −0.158* | −0.503* | (0.94) |

**Note.** *p < .05, p < .01. n = 200; values in parentheses represent Cronbach's alpha coefficients, values in parentheses indicate Cronbach's alpha coefficients. QoR-40 = Quality of Recovery-40; RHODES = Rhodes Index of Nausea, Vomiting, and Retching; SD = standard deviation; BMI = body mass index; VAS = Visual Analog Scale.

mean score for the *Pain* dimension was 19.17 (SD = 0.63), indicating that patients generally experienced low-to-moderate levels of pain. The mean score of the Rhodes Index was 15.13 (SD = 0.84), suggesting that patients experienced nausea, vomiting, and retching symptoms at a mild-to-moderate level. These findings indicate that although patients undergoing laparoscopic cholecystectomy generally reported a moderate-to-high recovery experience, symptoms such as nausea, vomiting, and pain had a marked impact on their overall recovery quality.

Regarding reliability analyses, the Cronbach's alpha coefficient for the overall Quality of Recovery Scale was .90, indicating a high level of internal consistency. Subscales demonstrated high reliability for *Physical Comfort* (.83), *Emotional State* (.78), and *Patient Support* (.87), while *Independence* (.60) and *Pain* (.66) demonstrated acceptable reliability. For the Rhodes Index, the Cronbach's alpha coefficient was calculated as .94, reflecting excellent internal consistency.

Pearson correlation coefficients revealed a significant and strong negative relationship between overall quality of recovery and the Rhodes Index (r = −.635, p < .01). This finding indicates that as recovery quality increases, nausea, vomiting, and retching symptoms decrease. Similarly, strong negative correlations were identified between the Rhodes Index and the subscales of *Physical Comfort* (r = −.777), *Emotional State* (r = −.619), and *Pain* (r = −.503). These results suggest that physical and psychological well-being are inversely related to physiological symptoms, indicating that reductions in nausea and vomiting are associated with improvements in the core components of recovery quality.

### 3.4. Findings Related to the Research Hypotheses

In line with the primary aim of this study, patients' levels of nausea and vomiting were identified as the independent variable (X), while recovery quality was defined as the dependent variable (Y). Within the relationship between nausea–vomiting and recovery quality in patients undergoing laparoscopic cholecystectomy, sociodemographic characteristics (e.g., gender, age, smoking status, body weight status) and pain levels were examined as moderating variables (W). The purpose was to investigate the potential moderating roles of these variables in the interaction between nausea vomiting and recovery quality. The results of the simple moderating analysis conducted to test the research hypotheses are presented in Table 5.

The results of the moderating analysis revealed that nausea and vomiting, as measured by the Rhodes Index, were significantly and negatively associated with recovery quality. Several sociodemographic and clinical factors moderating this relationship. Specifically, gender, body mass index (BMI), income level, smoking status, smoking frequency, chronic disease status, and postoperative pain intensity (VAS scores) exerted significant moderating effects (H1, H3, H6, H7, H8, H10, H12). These findings suggest that the negative impact of nausea and vomiting on recovery quality varies across patient subgroups defined by these characteristics. In contrast, age, marital status, educational level, alcohol consumption, and history of previous surgery did not demonstrate significant moderating roles (H2, H4, H5, H9, H11).

**Table 5. Findings of the Simple Moderating Analysis Related to the Research Hypotheses (Model 1).**

| Hypotheses and Independent Variable | B | SE | t | p | LLCI | ULCI | R² (Model) | Interaction Significance | Hypothesis Outcome |
|---|---|---|---|---|---|---|---|---|---|
| H1 Constant | 4.205 | 0.025 | 166.07 | 0.000 | 4.156 | 4.255 | 0.439 | ✓ | Accepted |
| Rhodes Nausea, Vomiting and Retching (X) | −0.360 | 0.030 | −11.84 | 0.000 | −0.420 | −0.300 | 0.439 | | |
| Gender (W) | 0.118 | 0.052 | 2.240 | 0.026 | 0.013 | 0.221 | 0.439 | | |
| X × Gender (Interaction) | −0.168 | 0.064 | −2.60 | 0.010 | −0.296 | −0.040 | 0.439 | | |
| H2 Constant | 4.211 | 0.026 | 161.55 | 0.000 | 4.160 | 4.262 | 0.403 | X | Rejected |
| Rhodes (X) | −0.358 | 0.031 | −11.46 | 0.000 | −0.420 | −0.297 | 0.403 | | |
| Age (W) | −0.000 | 0.002 | −0.034 | 0.972 | −0.004 | 0.004 | 0.403 | | |
| X × Age (Interaction) | 0.000 | 0.002 | 0.104 | 0.917 | −0.005 | 0.005 | 0.403 | | |
| H3 Constant | 4.609 | 0.363 | 12.675 | 0.000 | 3.892 | 5.327 | 0.406 | ✓ | Accepted |
| Rhodes (X) | −0.277 | 0.173 | −1.598 | 0.001 | −0.620 | −0.060 | 0.406 | | |
| BMI (W) | 0.009 | 0.012 | 0.787 | 0.032 | −0.014 | −0.004 | 0.406 | | |
| X × BMI (Interaction) | −0.002 | 0.005 | −0.482 | 0.030 | −0.014 | −0.008 | 0.406 | | |
| H4 Constant | 4.211 | 0.026 | 162.02 | 0.000 | 4.160 | 4.262 | 0.405 | X | Rejected |
| Rhodes (X) | −0.357 | 0.031 | −11.45 | 0.000 | −0.419 | −0.296 | 0.405 | | |
| Marital Status (W) | 0.055 | 0.081 | 0.676 | 0.499 | −0.105 | 0.216 | 0.405 | | |
| X × Marital Status (Interaction) | −0.031 | 0.088 | −0.354 | 0.723 | −0.205 | 0.143 | 0.405 | | |
| H5 Constant | 4.210 | 0.025 | 162.71 | 0.000 | 4.159 | 4.266 | 0.413 | X | Rejected |
| Rhodes (X) | −0.355 | 0.031 | −11.46 | 0.000 | −0.416 | −0.294 | 0.413 | | |
| Education (W) | 0.034 | 0.019 | 1.764 | 0.079 | −0.004 | 0.074 | 0.413 | | |
| X × Education (Interaction) | −0.011 | 0.023 | −0.498 | 0.619 | −0.057 | 0.034 | 0.413 | | |
| H6 Constant | 4.208 | 0.025 | 162.39 | 0.000 | 4.157 | 4.259 | 0.413 | ✓ | Accepted |
| Rhodes (X) | −0.352 | 0.031 | −11.34 | 0.000 | −0.414 | −0.291 | 0.413 | | |
| Income (W) | 0.053 | 0.045 | 1.165 | 0.045 | −0.037 | −0.024 | 0.413 | | |
| X × Income (Interaction) | −0.074 | 0.055 | −1.355 | 0.037 | −0.184 | −0.034 | 0.413 | | |
| H7 Constant | 4.210 | 0.026 | 162.01 | 0.000 | 4.159 | 4.262 | 0.406 | ✓ | Accepted |
| Rhodes (X) | −0.360 | 0.031 | −11.56 | 0.000 | −0.421 | −0.298 | 0.406 | | |
| Smoking (W) | 0.015 | 0.054 | 0.286 | 0.035 | −0.091 | −0.023 | 0.406 | | |
| X × Smoking (Interaction) | −0.057 | 0.065 | 0.865 | 0.028 | −0.072 | −0.036 | 0.406 | | |
| H8 Constant | 4.234 | 0.041 | 101.94 | 0.000 | 4.151 | 4.317 | 0.502 | ✓ | Accepted |
| Rhodes (X) | −0.388 | 0.050 | −7.688 | 0.000 | −0.489 | −0.287 | 0.502 | | |
| Cigarette Consumption (W) | −0.026 | 0.050 | −0.536 | 0.023 | −0.127 | −0.073 | 0.502 | | |
| X × Cigarette Consumption (Interaction) | −0.041 | 0.065 | −0.633 | 0.019 | −0.171 | −0.038 | 0.502 | | |
| H9 Constant | 4.215 | 0.026 | 161.41 | 0.000 | 4.163 | 4.266 | 0.409 | X | Rejected |
| Rhodes (X) | −0.362 | 0.031 | −11.59 | 0.000 | −0.424 | −0.300 | 0.409 | | |
| Alcohol (W) | −0.120 | 0.127 | −0.946 | 0.345 | −0.372 | 0.131 | 0.409 | | |
| X × Alcohol (Interaction) | 0.193 | 0.151 | 1.274 | 0.204 | −0.105 | 0.492 | 0.409 | | |
| H10 Constant | 4.212 | 0.026 | 161.19 | 0.000 | 4.160 | 4.264 | 0.404 | ✓ | Accepted |
| Rhodes (X) | −0.359 | 0.031 | −11.49 | 0.000 | −0.421 | −0.298 | 0.404 | | |
| Chronic Disease (W) | 0.020 | 0.052 | 0.388 | 0.028 | −0.083 | −0.023 | 0.404 | | |
| X × Chronic Disease (Interaction) | −0.029 | 0.063 | −0.460 | 0.016 | −0.153 | −0.095 | 0.404 | | |
| H11 Constant | 4.211 | 0.026 | 161.78 | 0.000 | 4.159 | 4.262 | 0.403 | X | Rejected |
| Rhodes (X) | −0.359 | 0.031 | −11.46 | 0.000 | −0.420 | −0.297 | 0.403 | | |
| Surgery (W) | 0.006 | 0.060 | 0.109 | 0.913 | −0.112 | 0.126 | 0.403 | | |
| X × Surgery (Interaction) | 0.001 | 0.068 | 0.027 | 0.978 | −0.133 | 0.137 | 0.403 | | |

*(Continued)*

**Table 5.** (Continued)

| Hypotheses and Independent Variable | B | SE | t | p | LLCI | ULCI | R² (Model) | Interaction Significance | Hypothesis Outcome |
|---|---|---|---|---|---|---|---|---|---|
| **H12 Constant** | 4.169 | 0.026 | 157.47 | 0.000 | 4.117 | 4.222 | 0.582 | ✓ | Accepted |
| **Rhodes (X)** | −0.256 | 0.032 | −7.970 | 0.000 | −0.320 | −0.193 | 0.582 | | |
| **VAS (W)** | −0.306 | 0.051 | −5.976 | 0.000 | −0.407 | −0.205 | 0.582 | | |
| **X × VAS (Interaction)** | 0.169 | 0.061 | 2.748 | 0.007 | 0.047 | 0.291 | 0.582 | | |

**Note.** n = 200; B = Beta coefficient; SE = Standard Error; LLCI = Lower Level Confidence Interval (95%); ULCI = Upper Level Confidence Interval (95%); Bootstrap resampling method with 5000 samples was used.

## 4. Conclusions

Cholecystectomy remains the most commonly performed surgical procedure for the treatment of gallbladder diseases3. With the evolution of minimally invasive techniques, laparoscopic cholecystectomy has replaced open surgery as the gold standard and is now widely practiced [22]. However, postoperative complications such as pain, nausea, and vomiting are frequently observed after laparoscopic interventions and may adversely affect patients' recovery quality. The aim of this study was to examine the relationship between postoperative nausea and vomiting and recovery quality in patients undergoing laparoscopic cholecystectomy and to evaluate the moderating roles of sociodemographic factors and pain in this relationship.

### 4.1. Discussion of findings related to the sociodemographic and medical history characteristics of patients

When examining the sociodemographic characteristics of the patients, it was found that the majority were aged 40 and over, with a predominance of female patients and a high proportion being married. Regarding educational status, a significant portion of the patients were primary school graduates. In terms of economic status, most participants reported that their income was equal to their expenses. Based on Body Mass Index (BMI) data, it was determined that patients were generally classified as "overweight." Regarding substance use, most patients were non-smokers and did not consume alcohol. From the perspective of medical history, the majority of patients did not report any chronic illnesses, although a large portion had undergone at least one previous surgical procedure.

In the present study, most patients were middle-aged or older and predominantly female, which is consistent with previous reports indicating that laparoscopic cholecystectomy is more frequently performed in older adults and women [5,23–27]. The predominance of female patients is in line with evidence showing a higher prevalence of gallstone disease among women, likely related to hormonal factors [5,23–27]. The mean BMI of the sample indicated that most participants were overweight, consistent with earlier studies and with evidence linking higher BMI to increased gallstone risk [23,28,29]. The high proportion of married patients and those with low educational attainment reflects the sociodemographic profile reported in similar Turkish samples [4,14,30,31]. In line with previous studies, most participants were non-smokers and non-alcohol users [14,23,31], and the majority had no chronic disease [23,24], which may be related to the elective nature of laparoscopic cholecystectomy. A substantial proportion of patients reported a history of previous surgery, consistent with earlier findings [19,30,32].

### 4.2. Discussion of correlation analyses regarding the measurement tools

The findings of this study revealed a strong and statistically significant negative correlation between the total scores of the Quality of Recovery Scale and the Rhodes Index of Nausea, Vomiting, and Retching.This result indicates that as patients' complaints of nausea, vomiting, and retching increase, their perceived quality of recovery significantly

decreases. This finding is consistent with existing literature. For example, Schwartz and Gan (2020) reported similar outcomes, while Spaniolas et al. (2020), in a randomized controlled trial, demonstrated that a multimodal intervention protocol aimed at preventing postoperative nausea and vomiting significantly improved patients' recovery quality [33,34]. Similarly, Salamah et al. (2022) found that effectively managing nausea and vomiting in patients undergoing laparoscopic surgery not only enhanced physical comfort but also accelerated the recovery process and positively impacted overall recovery quality [35].

Specifically, following laparoscopic cholecystectomy, symptoms such as pain, nausea, and vomiting may arise due to physiological mechanisms including $CO_2$ insufflation into the abdominal cavity, diaphragmatic irritation, increased splanchnic pressure, and vagal nerve stimulation [7,15].The emergence of these symptoms can significantly compromise patients' physiological comfort and negatively influence the quality of recovery. Furthermore, these symptoms may not only lead to physical discomfort but also hinder oral intake, mobility, and medication adherence, ultimately delaying the overall recovery process. In patients experiencing severe nausea and vomiting, increased levels of anxiety, fatigue, and dissatisfaction have also been observed, which may contribute to prolonged hospital stays and increased care needs. Therefore, effective management of such symptoms is considered crucial for enhancing not only physiological comfort but also psychological well-being, thereby contributing multidimensionally to the quality of recovery.

At the subscale level, correlation analyses revealed statistically significant negative relationships between the Rhodes Scale scores and the physical comfort, emotional state, and pain status subscales of the Quality of Recovery Scale. These results suggest that postoperative symptoms such as nausea and vomiting are inversely related to both physiological and psychological well-being. The obtained correlations support the notion that symptom management is a key determinant of postoperative recovery quality. This is consistent with findings in the literature. For instance, in a study conducted by Markos et al. (2024), a history of nausea and vomiting was significantly associated with lower recovery quality 11. Similarly, Schwartz and Gan (2020) reported that physiological symptoms such as nausea and vomiting not only reduced physical comfort but also adversely affected overall patient well-being, ultimately diminishing recovery quality [33].

Collectively, these findings indicate that common postoperative symptoms such as nausea and vomiting impact recovery quality not only physiologically but also psychologically, underscoring their multidimensional effects. Therefore, symptom management should be considered a primary area of intervention in perioperative nursing care processes. Moreover, symptom control strategies should extend beyond pharmacological treatments to include non-pharmacological nursing interventions, which are also likely to have a positive impact on recovery outcomes.

Supporting the effectiveness of non-pharmacological nursing interventions, a systematic review by Arslan and Şenol Çelik (2024) reported that interventions such as acupuncture, aromatherapy, oral ginger intake, music therapy, patient education, and patient visits were effective in reducing postoperative nausea and vomiting [36]. The reduction in these symptoms, in turn, positively influenced patients' perceptions of their recovery. In this regard, holistic nursing approaches to symptom management are considered fundamental components in enhancing postoperative recovery quality.

### 4.3. Discussion of findings related to the quality of recovery-40 (QoR-40) Scale Scores

Postoperative recovery quality is considered a critical indicator for evaluating patients' health status following surgery (Günel, 2021). Numerous factors influence recovery quality, including the patient's age, preoperative health status, type and duration of surgical intervention and anesthesia, presence of chronic diseases, nutritional status in elderly patients, and postoperative complications such as pain, nausea, vomiting, hypoglycemia, hyperglycemia, and insulin resistance. It is essential to assess each of these factors when evaluating recovery quality [37].

In the present study, the mean total score of the QoR-40 was found to be 143.18, indicating that patients generally experienced a moderating -to-high level of recovery quality. Similar findings have been reported in studies by Çetin (2024) and Okcul (2022), who found high levels of recovery quality among patients undergoing cholecystectomy. These

consistent results in the literature support the findings of the present research. The data suggest that patients undergoing laparoscopic cholecystectomy are generally satisfied with postoperative care and perceive their recovery positively. This may be attributed to the benefits of modern surgical techniques, effective symptom and pain management, high-quality nursing care, and comprehensive patient education.

When examining the subdimensions of the QoR-40, the highest mean score was observed in the "Physical Comfort" subscale. This aligns with the findings of Çetin (2024) in patients undergoing laparoscopic cholecystectomy, as well as Kavrazlı (2015), who reported similar results in patients from different surgical specialties (cardiovascular, general, and orthopedic surgery). These findings collectively suggest that physical comfort is a key determinant of recovery quality following surgery. The high score in this dimension may be explained by the minimally invasive nature of laparoscopic cholecystectomy, which causes less physical trauma compared to open surgery. As a result, physical comfort is preserved, and essential physiological processes such as sleep, respiration, and nutrition are largely maintained, contributing to better postoperative outcomes.

### 4.4. Discussion of findings related to the rhodes index of nausea, vomiting, and retching

Postoperative nausea and vomiting are among the most common complications following laparoscopic cholecystectomy. These symptoms may arise due to factors such as pneumoperitoneum, general anesthesia, manipulation of intra-abdominal organs, and postoperative analgesic use [38].The use of combination antiemetic therapy, involving medications with different mechanisms of action such as ondansetron, droperidol, and dexamethasone, has been found to be more effective than monotherapy in reducing the risk of postoperative nausea and vomiting [39].

In the present study, the mean score on the Rhodes Index was 15.13, indicating that patients generally experienced mild to moderating levels of nausea, vomiting, and retching in the postoperative period. The literature supports this finding. For example, Jasem and Dawood (2024) also reported moderating postoperative nausea and vomiting among patients after surgery [40]. Similarly, Doğan Kırtıloğu (2023) observed mild-to- moderating nausea in patients following laparoscopic cholecystectomy. These findings are consistent with those of the current study and underscore the importance of clinical monitoring of postoperative nausea and vomiting as a routine part of postoperative care.

The frequency and severity of these symptoms may be associated with the gastrointestinal effects of surgical techniques and medications used during the procedure. In a meta-analysis conducted by Amirshahi et al. (2020), which reviewed 23 studies involving 22,683 patients from various countries, nausea and vomiting were found to occur at high rates following surgery2. Similarly, in a study by Yayla et al. (2022), predictive factors for postoperative nausea and vomiting after laparoscopic cholecystectomy were identified, and high levels of nausea and vomiting were reported14.

Overall, the findings of this study regarding nausea and vomiting are largely consistent with the existing literature. It is suggested that the pneumoperitoneum created during laparoscopic cholecystectomy, by increasing intra-abdominal pressure, may temporarily impair gastrointestinal physiology. This may lead to suppressed gastric motility and diaphragmatic irritation [41]. Diaphragmatic irritation can stimulate the vomiting center in the brainstem via the vagus nerve, contributing to the onset of these symptoms [42].

Therefore, it is plausible that the physiological changes specific to laparoscopic surgery contribute to the higher incidence and severity of postoperative nausea and vomiting.

### 4.5. Discussion of findings related to the research hypothesis

**H1: Gender moderating the relationship between nausea, vomiting, and retching levels and quality of recovery in patients undergoing laparoscopic cholecystectomy. (Accepted)**

The results of this study revealed that gender had a statistically significant and positive direct effect on recovery quality. The proposed model explained 43.9% of the total variance, indicating that female patients reported lower recovery quality compared to male patients. Additionally, as scores on the Rhodes Index of Nausea, Vomiting, and Retching increased,

a decrease in recovery quality was observed in both genders. At the interaction level, an additional 1.93% increase in variance was detected, which was also statistically significant, This finding confirms that gender significantly moderates the relationship between nausea, vomiting, and retching symptoms and recovery quality. Notably, the decline in recovery quality associated with these symptoms was more pronounced among female patients, suggesting that women's perceptions of recovery are more adversely affected by such symptoms than those of men. Thus, **H1 was supported.**

Supporting literature also highlights similar patterns. In a study by Qian et al. (2022) involving 1,670 outpatients, postoperative nausea and vomiting were observed in 156 cases, 112 of whom were female, indicating a higher prevalence among women [43]. Similarly, in the study by Yayla et al. (2022) with 172 laparoscopic cholecystectomy patients, nausea and vomiting scores were found to be higher in women than in men14. Salazar-Parra et al. (2020) also reported that early postoperative symptoms such as pain, nausea, and vomiting were more prevalent in female patients after elective laparoscopic cholecystectomy, and that women required more analgesic and antiemetic medications [44].

These findings are largely consistent with the present study, suggesting that female patients are more sensitive to postoperative symptoms such as nausea and vomiting, which in turn more negatively affect their perceived recovery quality. Confessor de Sousa et al. (2025), in their evaluation of recovery quality following video-laparoscopic cholecystectomy, also identified gender as a significant predictor, with women reporting lower recovery quality [45]. Similarly, in a study by Çetin (2024) evaluating recovery over three postoperative days, women had consistently lower QoR scores than men on all days. Dığın and Özkan (2021) also found lower postoperative recovery scores in female patients [46].

The results of this study, in agreement with the literature, indicate that women are more susceptible to postoperative symptoms such as nausea and vomiting and that these symptoms more substantially impair their recovery experience. Several multidimensional factors may explain this outcome. Prior research suggests that women have greater sensitivity to symptoms, anticipate more discomfort during the postoperative period, and evaluate their symptoms more negatively than men [47]. Furthermore, the higher incidence of nausea and vomiting in women may be related to estrogen-induced sensitization of the chemoreceptor trigger zone and the vomiting center [48]. Thus, the increased prevalence and impact of these symptoms in women may stem from both physiological interactions involving estrogen and individual symptom perception differences.

**H3: Body Mass Index (BMI) moderating the relationship between nausea, vomiting, and retching levels and quality of recovery in patients undergoing laparoscopic cholecystectomy. (Accepted)**

Postoperative symptoms such as nausea, vomiting, and pain can directly influence the overall recovery process in patients after laparoscopic cholecystectomy. The severity of these symptoms may vary depending on patients' physiological structure, preoperative health status, and BMI. In this study, the mean BMI of patients was found to be 28.89 ± 3.214, placing the majority of participants in the overweight category.

When analyzing the moderating role of BMI distributions, the proposed model was found to be statistically significant, explaining 40.6% of the variance in recovery quality. The interaction term between the Rhodes Index of Nausea, Vomiting, and Retching and BMI was statistically significant. These results indicate that increases in nausea, vomiting, and retching symptoms are associated with decreased recovery quality. However, the strength of this relationship varies across different BMI levels.

Among patients with low BMI, the decline in recovery quality associated with increasing nausea and vomiting was more gradual and less severe. In contrast, patients in the moderating moderating BMI category experienced a more pronounced decline in recovery quality as symptom severity increased. Most notably, patients with high BMI exhibited a steep and statistically significant decline in recovery quality in response to increased symptoms.

These findings suggest that as BMI increases, patients may become more physiologically sensitive to symptoms such as nausea and vomiting, and these symptoms have a more substantial negative impact on their perceived recovery quality. Furthermore, BMI itself had a significant and positive main effect on recovery quality, reinforcing its role as a meaningful predictor in postoperative recovery outcomes.

In light of these findings, **Hypothesis H3 was supported**, indicating that BMI plays a significant moderating role in the relationship between postoperative nausea, vomiting, and recovery quality. These results underscore the importance of individualized postoperative care that considers patients' BMI in managing symptoms and enhancing recovery experiences.

**H6: Income level moderating the relationship between nausea, vomiting, and retching levels and quality of recovery in patients undergoing laparoscopic cholecystectomy. (Accepted)**

In this study, the direct effect of income level on quality of recovery was found to be positive but marginally significant, suggesting that income alone may have a limited yet noteworthy influence on patients' perceptions of recovery quality. The most striking finding, however, lies in the significant interaction effect between the Rhodes Index of Nausea, Vomiting, and Retching and income level. This interaction increased the model's explanatory power by 0.55%.

According to the results, while recovery quality decreased as nausea, vomiting, and retching symptoms increased across all income groups, the *magnitude* of this decline varied depending on income status. Specifically, among patients with higher income levels, the negative impact of Rhodes scores on recovery quality was more attenuated. In contrast, among patients whose income was lower than their expenses, the decline in recovery quality was steeper and more pronounced. These findings suggest that patients with higher income may be more resilient to postoperative physical discomforts, or may have better access to resources that help mitigate the effects of such symptoms.

Therefore, **Hypothesis H6 was supported**, confirming that income level plays a moderating role in the relationship between postoperative symptom severity and perceived recovery quality.

The findings are supported by previous studies in the literature. For instance, in a retrospective analysis by Lee et al. (2023), socioeconomic factors were shown to influence both the severity of postoperative nausea and vomiting and adherence to antiemetic prophylaxis protocols [49].Similarly, Beswick et al. (2019) found that patients with higher income levels reported significantly better recovery outcomes following surgery [50]. Jager, Gunnarsson, and Ho (2022) also observed that patients with lower socioeconomic status were at greater risk for postoperative complications and faced more difficulties during recovery [51].

Although these prior studies included broader surgical populations, the consistency of the findings with the present study focusing specifically on laparoscopic cholecystectomy patients suggests that the effect of socioeconomic status on recovery quality is not limited to a specific surgical type.

Ultimately, the current study indicates that the negative impact of nausea, vomiting, and retching symptoms on recovery is less severe in patients with higher income levels. This supports the notion that socioeconomic status has both direct and indirect effects on recovery quality, and that patients from lower-income groups may experience greater challenges in the postoperative period. These findings highlight the potential influence of economic conditions on both the experience and management of postoperative symptoms such as nausea and vomiting.

**H7: Smoking status moderating the relationship between nausea, vomiting, and retching levels and quality of recovery in patients undergoing laparoscopic cholecystectomy. (Accepted)**

Smoking status appears to differentially shape the relationship between postoperative nausea and vomiting and perceived recovery quality. Consistent with previous reports, non-smokers tend to experience higher rates of postoperative nausea and vomiting than smokers following laparoscopic cholecystectomy [14,38] and other surgical procedures [52,53]. Although smokers may demonstrate attenuated symptom perception, potentially related to altered physiological responses to emetogenic and anesthetic agents [53,55], smoking is consistently associated with less favorable postoperative outcomes, including lower recovery quality and higher complication rates [54]. From a nursing perspective, these findings underscore the importance of comprehensive symptom assessment and targeted postoperative care that accounts for smoking status, while emphasizing smoking cessation and evidence-based supportive interventions to optimize recovery.

**H8: The amount of cigarette consumption moderating the relationship between nausea, vomiting, and retching levels and quality of recovery in patients undergoing laparoscopic cholecystectomy. (Accepted)**

The amount of daily cigarette consumption appears to differentially influence postoperative recovery quality by shaping how nausea and vomiting symptoms are experienced and translated into recovery outcomes. In contrast to earlier studies suggesting a possible tolerance effect among heavier smokers [52,56], the present findings indicate that greater smoking intensity may be associated with a more pronounced deterioration in perceived recovery when postoperative symptoms are present. Consistent with the broader literature linking smoking to poorer postoperative outcomes and higher complication risk [54], these results highlight that smoking intensity, beyond smoking status alone, is a clinically relevant factor in postoperative recovery. From a nursing perspective, systematic assessment of smoking behavior and the integration of preoperative smoking reduction or cessation counseling into perioperative care pathways may support more targeted symptom management and improved recovery trajectories. **H10: The presence of chronic illness moderating the relationship between nausea, vomiting, and retching levels and quality of recovery in patients undergoing laparoscopic cholecystectomy. (Accepted)**

This study investigated the moderating role of chronic illness on recovery quality in the postoperative period. The findings revealed that chronic illness had a statistically significant direct effect on recovery quality. Additionally, the interaction between the Rhodes Index of Nausea, Vomiting, and Retching and chronic illness status was also statistically significant, contributing an additional 0.06% to the model's explanatory power.

These results indicate that patients with chronic illnesses experience significantly more difficulty in postoperative recovery. As nausea and vomiting symptoms increased, quality of recovery decreased across both groups. However, this decline was more pronounced among patients with chronic conditions. This suggests that individuals with chronic illnesses are more vulnerable to the physiological and psychological effects of postoperative symptoms, resulting in a greater deterioration in their perceived recovery.

Therefore, **Hypothesis H10 was supported**, showing that chronic illness status is a significant moderating factor in the relationship between symptom severity and recovery quality.

The literature supports these findings. For example, Woudneh (2025) found that patients with chronic conditions such as diabetes and hypertension exhibited poorer recovery outcomes after surgery, including those who underwent laparoscopic cholecystectomy [57]. Similarly, Markos et al. (2024) reported that pain intensity, nausea and vomiting, and coexisting chronic diseases directly affected postoperative recovery quality11. Özman (2024) found that patients without chronic conditions had significantly higher recovery scores and experienced more favorable postoperative outcomes. Fowler et al. (2022) reported that one in four surgical patients had at least one chronic condition, and these patients demonstrated lower recovery quality in the postoperative period [58].

Collectively, these findings suggest that chronic diseases can negatively impact recovery quality by impairing wound healing, suppressing immune responses, and disrupting metabolic balance. These biological disruptions likely make patients more susceptible to complications and slower recovery.

Consistent with these studies, the current research demonstrated that patients with chronic diseases were more severely affected by postoperative symptoms and had notably lower recovery scores compared to those without such conditions. Symptoms like vomiting and retching may trigger further complications in these patients. For instance, fluctuations in blood pressure, dehydration due to fluid loss, and electrolyte imbalances are more likely to develop in patients with chronic illnesses. This increased risk contributes to a stronger perception of discomfort and a more significant decline in overall recovery quality.

In conclusion, **chronic illness plays a meaningful moderating role** in the relationship between postoperative symptoms—especially nausea, vomiting, and pain and recovery quality. These patients require more comprehensive monitoring and supportive care to mitigate symptom severity and optimize postoperative recovery outcomes.

**H12: Pain intensity (VAS scores) moderating the relationship between nausea, vomiting, and retching levels and quality of recovery in patients undergoing laparoscopic cholecystectomy. (Accepted)**

In this study, the direct effect of pain severity measured by Visual Analog Scale (VAS) scores on recovery quality was found to be statistically significant Moreover, the interaction between the Rhodes Index of Nausea, Vomiting, and Retching

and VAS scores was also significant, indicating that pain not only directly influences recovery quality but also moderating the impact of postoperative symptoms such as nausea and vomiting on recovery outcomes.

This interaction improved the explanatory power of the model by 1.61%, underscoring the moderating role of pain in shaping how postoperative symptoms affect recovery. As Rhodes scores increased, recovery quality decreased across all pain levels; however, the **magnitude** of this decrease varied by pain intensity.

Among patients in the **low-pain group** (VAS 0–2), the impact of nausea and vomiting on recovery quality was present but limited. In contrast, patients in the **high-pain group** (VAS 6+) experienced a substantially greater decline in recovery quality as symptom severity increased. These findings clearly support **Hypothesis H12**, identifying pain as a critical factor that exacerbates the negative influence of nausea and vomiting on postoperative recovery.

The results align with prior research. For instance, Yayla et al. (2022) found that patients with higher postoperative pain scores also reported higher levels of nausea and vomiting after laparoscopic cholecystectomy14. Similarly, Sarin, Urman, and Ohno-Machado (2012) observed that increasing postoperative pain intensity was associated with a rise in nausea and vomiting symptoms [59].

Further supporting evidence comes from Markos et al. (2024), who reported that patients with a VAS score ≥ 7 and a history of nausea and vomiting were significantly more likely to experience poor recovery outcomes11. Their study found that patients with severe postoperative pain had **twice the risk** of poor recovery compared to those with mild or moderate pain. Additionally, those experiencing nausea and vomiting had a **2.5 times greater risk** of low recovery quality. These findings are consistent with the current study's results and emphasize that **both pain management and symptom control** are essential for maintaining a high quality of recovery after surgery.

In conclusion, postoperative pain significantly influences the relationship between gastrointestinal symptoms and recovery, with higher pain levels amplifying the detrimental effects of nausea and vomiting on patient outcomes. These findings highlight the need for **multimodal pain and symptom management protocols** to support optimal recovery following laparoscopic cholecystectomy.

### 4.6. Implications for clinical practice and nursing care

The findings highlight the importance of systematic assessment of postoperative nausea, vomiting, and pain as core components of nursing care in patients undergoing laparoscopic cholecystectomy. Identifying patients who are more vulnerable to symptom-related deterioration in recovery quality based on pain levels and selected sociodemographic characteristics may support more individualized and proactive nursing interventions. Integrating targeted symptom management strategies into routine postoperative care planning may contribute to improved recovery experiences and patient-centered outcomes. In addition, incorporating brief preoperative counseling on modifiable risk factors and reinforcing postoperative monitoring of symptom burden may strengthen the quality and effectiveness of perioperative nursing care.

### 4.7. Limitations of the Work

This study has several limitations. The cross-sectional design limits causal interpretation. The use of self-report measures may introduce response bias. Multiple moderator analyses may increase the risk of Type I error. All variables were assessed at a single postoperative time point. Although multi-center, the study was conducted within a single national healthcare system, which may limit generalizability. Future longitudinal studies are recommended to validate these findings.

### 5. Conclusion and Recommendations

This study investigated the impact of postoperative nausea, vomiting, and pain on the quality of recovery in patients undergoing laparoscopic cholecystectomy, as well as the moderating roles of sociodemographic variables and pain intensity. The findings indicated that as symptom severity increased, recovery quality significantly decreased. Notably, female

gender, high BMI, low income, smoking status, and the presence of chronic illness were identified as factors that intensified the negative impact of symptoms on recovery. Pain was found to affect recovery quality both directly and indirectly through its interaction with nausea and vomiting.

In light of these findings, the following recommendations are proposed to improve recovery outcomes after laparoscopic cholecystectomy:

• Development of individualized symptom management protocols specifically targeting pain, nausea, and vomiting;

• Implementation of structured nursing follow-up and monitoring systems for high-risk patient groups;

• Provision of in-service training programs to enhance nurses' competencies in symptom management and patient education;

• Conduct of advanced nursing research focusing on long-term recovery outcomes and the development of evidence-based interventions.

## Supporting information

**S1 Data. Data.**
(XLSX)

## Acknowledgements

Informed Consent: Written informed consent was obtained from all participantswho participated in this study.

## Author contributions

**Conceptualization:** İlknur Taşçi, Gizem Kubat Bakir.

**Data curation:** İlknur Taşçi.

**Formal analysis:** İlknur Taşçi, Gizem Kubat Bakir.

**Investigation:** İlknur Taşçi, Gizem Kubat Bakir.

**Methodology:** İlknur Taşçi, Gizem Kubat Bakir.

**Resources:** İlknur Taşçi, Gizem Kubat Bakir.

**Software:** İlknur Taşçi, Gizem Kubat Bakir.

**Validation:** İlknur Taşçi, Gizem Kubat Bakir.

**Visualization:** İlknur Taşçi, Gizem Kubat Bakir.

**Writing – original draft:** İlknur Taşçi, Gizem Kubat Bakir.

**Writing – review & editing:** Gizem Kubat Bakir.

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
