## [Decision Letter · Decision Letter 0]

4 Feb 2026

PONE-D-25-63021Pain and Sociodemographic Factors as Mediating Roles in the Relationship Between Postoperative Nausea–Vomiting and Recovery Quality: A PROCESS Macro Modeling Study From a Nursing Perspective on Laparoscopic Cholecystectomy PatientsPLOS One

Dear Dr. KUBAT BAKIR,

Thank you for submitting your manuscript to PLOS ONE. After careful consideration, we feel that it has merit but does not fully meet PLOS ONE’s publication criteria as it currently stands. Therefore, we invite you to submit a revised version of the manuscript that addresses the points raised during the review process.

Revise.

We look forward to receiving your revised manuscript.

Kind regards,

Robert Jeenchen Chen, MD, MPH, ChFC®, EA

Academic Editor

PLOS One

2. Peer review at PLOS One is not double-blinded (https://journals.plos.org/plosone/s/editorial-and-peer-review-process). For this reason, authors should include in the revised manuscript all the information removed for blind review.

3. In the online submission form you indicate that your data is not available for proprietary reasons and have provided a contact point for accessing this data. Please note that your current contact point is a co-author on this manuscript. According to our Data Policy, the contact point must not be an author on the manuscript and must be an institutional contact, ideally not an individual. Please revise your data statement to a non-author institutional point of contact, such as a data access or ethics committee, and send this to us via return email. Please also include contact information for the third party organization, and please include the full citation of where the data can be found.

Reviewers' comments:

Reviewer's Responses to Questions

**Comments to the Author**

1. Is the manuscript technically sound, and do the data support the conclusions?

Reviewer #1: Yes

Reviewer #2: Yes

2. Has the statistical analysis been performed appropriately and rigorously? 

Reviewer #1: Yes

Reviewer #2: Yes

3. Have the authors made all data underlying the findings in their manuscript fully available?

The PLOS Data policy requires authors to make all data underlying the findings described in their manuscript fully available without restriction, with rare exception (please refer to the Data Availability Statement in the manuscript PDF file). The data should be provided as part of the manuscript or its supporting information, or deposited to a public repository. For example, in addition to summary statistics, the data points behind means, medians and variance measures should be available. If there are restrictions on publicly sharing data—e.g. participant privacy or use of data from a third party—those must be specified.requires authors to make all data underlying the findings described in their manuscript fully available without restriction, with rare exception (please refer to the Data Availability Statement in the manuscript PDF file). The data should be provided as part of the manuscript or its supporting information, or deposited to a public repository. For example, in addition to summary statistics, the data points behind means, medians and variance measures should be available. If there are restrictions on publicly sharing data—e.g. participant privacy or use of data from a third party—those must be specified.requires authors to make all data underlying the findings described in their manuscript fully available without restriction, with rare exception (please refer to the Data Availability Statement in the manuscript PDF file). The data should be provided as part of the manuscript or its supporting information, or deposited to a public repository. For example, in addition to summary statistics, the data points behind means, medians and variance measures should be available. If there are restrictions on publicly sharing data—e.g. participant privacy or use of data from a third party—those must be specified.requires authors to make all data underlying the findings described in their manuscript fully available without restriction, with rare exception (please refer to the Data Availability Statement in the manuscript PDF file). The data should be provided as part of the manuscript or its supporting information, or deposited to a public repository. For example, in addition to summary statistics, the data points behind means, medians and variance measures should be available. If there are restrictions on publicly sharing data—e.g. participant privacy or use of data from a third party—those must be specified.

Reviewer #1: Yes

Reviewer #2: Yes

4. Is the manuscript presented in an intelligible fashion and written in standard English?

Reviewer #1: Yes

Reviewer #2: Yes

5. Review Comments to the Author

Reviewer #1: The article will be ready for publication with minor corrections. My suggestions are as follows: 1. TITLE: Although the title uses the phrase "mediating roles," the analyses were conducted using PROCESS Model 1 (moderation). The concepts of mediation and moderation have been confused. The title should be corrected to "moderation."

2. ABSTRACT: The purpose and method are generally understandable. However: The phrase "Mediating effects" is used incorrectly again. Process Model 1 tests interaction (moderation), not mediation. I think the translation in the abstract section is incorrect. It should be corrected.

3. STATISTICAL ANALYSIS AND PROCESS USAGE: PROCESS Macro Model 1 is for moderation, not mediation. In the text, systematically: “mediating role” “mediation effect” the expressions are used incorrectly. This is a conceptual error and must be corrected.

4. LIMITATIONS: The limitations section is insufficient. The following must be added: Cross-sectional design Self-report scales Risk of multiple comparisons Single-time measurement

The limitations are scattered throughout the discussion. They should be grouped under a separate, concise paragraph titled “Limitations”.

5. DİSCUSSİON: The discussion section should be shortened by at least 30–40%, redundant literature reviews should be removed, and the comments should be simplified to focus on the study's main contribution.

Kind regards

Özen İNAM

Reviewer #2: This study addresses an important clinical issue and is generally well designed and clearly written. The manuscript is suitable for publication after minor revisions, as outlined below.

• The abstract contains an excessive amount of numerical detail. Focusing more on the main findings rather than specific statistics would improve clarity and readability.

• The introduction provides a solid overview of the topic; however, the unique contribution of the study could be stated more explicitly in the final paragraph.

• The research questions and/or hypotheses would benefit from being more clearly emphasized at the end of the introduction.

• The study design is appropriately described; however, it would be useful to briefly note that the cross-sectional nature of the study does not allow causal interpretations.

• In the statistical analysis section, the use of the PROCESS Macro could be supported with a brief methodological justification to enhance transparency.

• Table titles could be made more descriptive and informative.

• All abbreviations used in the tables should be fully explained in the table footnotes.

• The discussion section is comprehensive; however, some paragraphs could be shortened and reorganized to improve focus, and the use of short subheadings may enhance readability.

• The implications for clinical practice and nursing care should be highlighted more clearly, preferably in a dedicated paragraph.

• The discussion of sociodemographic characteristics may be reduced or removed from the discussion section, as these variables are already adequately presented in the results.

• The limitations section should explicitly acknowledge the cross-sectional design and the restriction of the sample to a specific geographic region, along with their potential impact on generalizability.

6. PLOS authors have the option to publish the peer review history of their article (what does this mean?). If published, this will include your full peer review and any attached files.). If published, this will include your full peer review and any attached files.). If published, this will include your full peer review and any attached files.). If published, this will include your full peer review and any attached files.

...

Reviewer #1: **Yes:** Özen İnamÖzen İnamÖzen İnamÖzen İnam

Reviewer #2: No

---

## [Author Response · Author response to Decision Letter 1]

9 Feb 2026

We would like to sincerely thank the reviewers for their careful reading of our manuscript and for their constructive and insightful comments. We believe that these suggestions have significantly strengthened the clarity, rigor, and overall quality of the manuscript. All comments have been addressed point by point in the revised version, and corresponding changes have been highlighted in the manuscript.

Reviewer #1

1. TITLE: Although the title uses the phrase "mediating roles," the analyses were conducted using PROCESS Model 1 (moderation). The concepts of mediation and moderation have been confused. The title should be corrected to "moderation."

Thank you for this important clarification. The title has been revised to replace “mediating roles” with “moderating roles” to accurately reflect the use of PROCESS Model 1 (moderation) in the analyses.

2. ABSTRACT: The purpose and method are generally understandable. However: The phrase "Mediating effects" is used incorrectly again. Process Model 1 tests interaction (moderation), not mediation. I think the translation in the abstract section is incorrect. It should be corrected.

Thank you for the helpful comment. The wording in the Abstract has been corrected by replacing “mediating effects” with “moderating effects,” as PROCESS Model 1 examines interaction (moderation) rather than mediation.

3. STATISTICAL ANALYSIS AND PROCESS USAGE: PROCESS Macro Model 1 is for moderation, not mediation. In the text, systematically: “mediating role” “mediation effect” the expressions are used incorrectly. This is a conceptual error and must be corrected.

Thank you for highlighting this conceptual issue. We agree that PROCESS Macro Model 1 tests moderation (interaction) rather than mediation. Accordingly, we have systematically revised the Statistical Analysis section and all related parts of the manuscript by replacing the incorrect terms (“mediating role,” “mediation effect,” etc.) with the correct moderation terminology and ensured consistency throughout the text.

4. LIMITATIONS: The limitations section is insufficient. The following must be added: Cross-sectional design Self-report scales Risk of multiple comparisons Single-time measurement

The limitations are scattered throughout the discussion. They should be grouped under a separate, concise paragraph titled “Limitations”. Thank you for the suggestion. We added a concise “Limitations” paragraph that explicitly addresses the cross-sectional design, self-report measures, multiple comparison risk, and single-time measurement, grouped under a separate subheading as recommended.

5. DİSCUSSİON: The discussion section should be shortened by at least 30–40%, redundant literature reviews should be removed, and the comments should be simplified to focus on the study's main contribution. Thank you for the constructive feedback. The Discussion section has been substantially shortened redundant literature reviews were removed, and the text was streamlined to emphasize the main contributions of the study.

Reviewer #2

The abstract contains an excessive amount of numerical detail. Focusing more on the main findings rather than specific statistics would improve clarity and readability.

Thank you for the suggestion. The Abstract has been revised to reduce numerical details and to emphasize the main findings for improved clarity and readability.

The introduction provides a solid overview of the topic; however, the unique contribution of the study could be stated more explicitly in the final paragraph.

Thank you for the constructive comment. The final paragraph of the Introduction has been revised to more explicitly state the unique contribution of the present study and its specific added value to the existing literature.

The research questions and/or hypotheses would benefit from being more clearly emphasized at the end of the introduction.

Thank you for this valuable suggestion. We revised the final paragraph of the Introduction to better emphasize the relevance and clinical importance of the research questions from a nursing perspective.

The study design is appropriately described; however, it would be useful to briefly note that the cross-sectional nature of the study does not allow causal interpretations.

“Given the cross-sectional design, causal inferences cannot be made from the observed associations.”

In the statistical analysis section, the use of the PROCESS Macro could be supported with a brief methodological justification to enhance transparency.

“The PROCESS Macro was used because it provides a robust and widely accepted framework for testing moderation effects in observational data through regression-based interaction models.”

Table titles could be made more descriptive and informative.

Thank you for this helpful suggestion. The table titles have been revised to be more descriptive and informative.

All abbreviations used in the tables should be fully explained in the table footnotes.

Thank you for this helpful comment. All abbreviations used in the tables have now been fully explained in the table footnotes.

The discussion section is comprehensive; however, some paragraphs could be shortened and reorganized to improve focus, and the use of short subheadings may enhance readability.

Thank you for this helpful suggestion. The Discussion section has been revised by shortening and reorganizing selected paragraphs to improve focus, and short subheadings were added to enhance readability.

The implications for clinical practice and nursing care should be highlighted more clearly, preferably in a dedicated paragraph. Thank you for this suggestion. We added a dedicated paragraph highlighting the clinical and nursing practice implications of the findings in the Discussion section.

The discussion of sociodemographic characteristics may be reduced or removed from the discussion section, as these variables are already adequately presented in the results.

Thank you for this helpful suggestion. The discussion of sociodemographic characteristics has been substantially shortened and streamlined, with redundant descriptive content removed to avoid repetition of the Results section.

The limitations section should explicitly acknowledge the cross-sectional design and the restriction of the sample to a specific geographic region, along with their potential impact on generalizability.

Thank you for this helpful comment. We have revised the Limitations section to explicitly acknowledge the cross-sectional design and the restriction of the sample to a specific geographic region, and we noted their potential impact on generalizability.

---

## [Decision Letter · Decision Letter 1]

15 Feb 2026

Pain and Sociodemographic Factors as Mediating Roles in the Relationship Between Postoperative Nausea–Vomiting and Recovery Quality: A PROCESS Macro Modeling Study From a Nursing Perspective on Laparoscopic Cholecystectomy Patients

PONE-D-25-63021R1

Dear Dr. KUBAT BAKIR,

We’re pleased to inform you that your manuscript has been judged scientifically suitable for publication and will be formally accepted for publication once it meets all outstanding technical requirements.

Kind regards,

Robert Jeenchen Chen, MD, MPH, ChFC®, EA

Academic Editor

PLOS One

Additional Editor Comments (optional):

Reviewers' comments:

Reviewer's Responses to Questions

**Comments to the Author**

1. If the authors have adequately addressed your comments raised in a previous round of review and you feel that this manuscript is now acceptable for publication, you may indicate that here to bypass the “Comments to the Author” section, enter your conflict of interest statement in the “Confidential to Editor” section, and submit your "Accept" recommendation.

Reviewer #1: All comments have been addressed

Reviewer #2: All comments have been addressed

2. Is the manuscript technically sound, and do the data support the conclusions?

Reviewer #1: Yes

Reviewer #2: Yes

3. Has the statistical analysis been performed appropriately and rigorously? 

Reviewer #1: Yes

Reviewer #2: Yes

4. Have the authors made all data underlying the findings in their manuscript fully available?

The PLOS Data policy requires authors to make all data underlying the findings described in their manuscript fully available without restriction, with rare exception (please refer to the Data Availability Statement in the manuscript PDF file). The data should be provided as part of the manuscript or its supporting information, or deposited to a public repository. For example, in addition to summary statistics, the data points behind means, medians and variance measures should be available. If there are restrictions on publicly sharing data—e.g. participant privacy or use of data from a third party—those must be specified.requires authors to make all data underlying the findings described in their manuscript fully available without restriction, with rare exception (please refer to the Data Availability Statement in the manuscript PDF file). The data should be provided as part of the manuscript or its supporting information, or deposited to a public repository. For example, in addition to summary statistics, the data points behind means, medians and variance measures should be available. If there are restrictions on publicly sharing data—e.g. participant privacy or use of data from a third party—those must be specified.requires authors to make all data underlying the findings described in their manuscript fully available without restriction, with rare exception (please refer to the Data Availability Statement in the manuscript PDF file). The data should be provided as part of the manuscript or its supporting information, or deposited to a public repository. For example, in addition to summary statistics, the data points behind means, medians and variance measures should be available. If there are restrictions on publicly sharing data—e.g. participant privacy or use of data from a third party—those must be specified.requires authors to make all data underlying the findings described in their manuscript fully available without restriction, with rare exception (please refer to the Data Availability Statement in the manuscript PDF file). The data should be provided as part of the manuscript or its supporting information, or deposited to a public repository. For example, in addition to summary statistics, the data points behind means, medians and variance measures should be available. If there are restrictions on publicly sharing data—e.g. participant privacy or use of data from a third party—those must be specified.

Reviewer #1: Yes

Reviewer #2: Yes

5. Is the manuscript presented in an intelligible fashion and written in standard English?

Reviewer #1: Yes

Reviewer #2: Yes

6. Review Comments to the Author

Reviewer #1: Thank you for implementing all the suggested corrections without omission. I wish you continued success.

Reviewer #2: (No Response)

7. PLOS authors have the option to publish the peer review history of their article (what does this mean?). If published, this will include your full peer review and any attached files.). If published, this will include your full peer review and any attached files.). If published, this will include your full peer review and any attached files.). If published, this will include your full peer review and any attached files.

...

Reviewer #1: No

Reviewer #2: No

---

## [Editor Report · Acceptance letter]

PONE-D-25-63021R1

PLOS One

Dear Dr. KUBAT BAKIR,

I'm pleased to inform you that your manuscript has been deemed suitable for publication in PLOS One. Congratulations! Your manuscript is now being handed over to our production team.

Kind regards,

on behalf of

Dr. Robert Jeenchen Chen

Academic Editor

PLOS One